# Evaluation of immune and chemical precipitation methods for plasma exosome isolation

Tatiana Shtam[1,2,3]*, Vladimir Evtushenko[4], Roman Samsonov[2,4], Yana Zabrodskaya[1,3,5], Roman Kamyshinsky[3,6], Lidia Zabegina[2,5,7], Nikolay Verlov[1,3], Vladimir Burdakov[1,3], Luiza Garaeva[1,3,5], Maria Slyusarenko[2,7], Nadezhda Nikiforova[2,7], Andrey Konevega[1,3,5], Anastasia Malek[2,7]*

**1** Petersburg Nuclear Physics Institute named by B.P. Konstantinov of National Research Center "Kurchatov Institute", Gatchina, Russia, **2** N.N. Petrov National Medical Research Center of Oncology, St. Petersburg, Russia, **3** National Research Center "Kurchatov Institute", Moscow, Russia, **4** Russian Scientific Center of Radiology and Surgical Technologies named by academician A.M. Granov, St. Petersburg, Russia, **5** Peter the Great St. Petersburg Polytechnic University, St. Petersburg, Russia, **6** Shubnikov Institute of Crystallography of Federal Scientific Research Centre "Crystallography and Photonics" of Russian Academy of Sciences, Moscow, Russia, **7** Ltd Oncosystem, Skolkovo Innovation Center, Moscow, Russia

* Shtam_TA@pnpi.nrcki.ru (TS); anastasia@malek.com (AM)

**Data Availability Statement:** All relevant data are within the paper and its Supporting information files.

## Abstract

Exosomes are a type of extracellular vesicles (EVs) secreted by multiple mammalian cell types and involved in intercellular communication. Numerous studies have explored the diagnostic and therapeutic potential of exosomes. The key challenge is the lack of efficient and standard techniques for isolation and downstream analysis of nanovesicles. Conventional isolation methods, such as ultracentrifugation, precipitation, filtration, chromatography, and immune-affinity-based approaches, rely on specific physical properties or on surface biomarkers. However, any of the existing methods has its limitations. Various parameters, such as efficacy, specificity, labor input, cost and scalability, and standardization options, must be considered for the correct choice of appropriate approach. The isolation of exosomes from biological fluids is especially challenged by the complex nature and variability of these liquids. Here, we present a comparison of five protocols for exosome isolation from human plasma: two chemical affinity precipitation methods (lectin-based purification and SubX™ technology), immunoaffinity precipitation, and reference ultracentrifugation-based exosome isolation method in two modifications. An approach for the isolation of exosomes based on the phenomenon of binding and aggregation of these particles via clusters of outer membrane phosphate groups in the presence of SubX™ molecules has been put forward in the present study. The isolated EVs were characterized based upon size, quantity, and protein content.

**Funding:** The study was supported by the Russian Science Foundation project 19-74-20146. The funders had no role in study design, data collection and analysis, decision to publish, or preparation of the manuscript.

**Competing interests:** The authors have declared that no competing interests exist.

## Introduction

Extracellular vesicles (EVs) are nanoscale size bubble-like membranous structures secreted by most cell types and present in blood, urine, saliva, breast milk and cerebrospinal fluid [1]. EVs contain lipids, proteins, RNA, metabolites and are thought to be involved in intercellular communication [2]. There are several categories of EVs: apoptotic bodies (500–1000 nm), which are released from cells undergoing apoptosis, microvesicles (100–350 nm), which are released by evagination of the plasma membrane, and exosomes (40–150 nm) [3]. Exosomes are a subtype of EVs that are released by the fusion of multivesicular bodies with the plasma membrane. Exosomes differ from other types of EVs by a relatively small size and the expression of specific exosomal markers (CD9, CD63, CD81 and others) [3,4].

Extracellular vesicles in general and especially exosomes play a role in cell-to-cell signaling and serve as possible biomarkers for disease diagnosis, prognosis, and therapy [5,6] Their pathophysiological roles are being decoded in various diseases including cancer. In addition, growing data also suggests that exosomes are involved in facilitating oncogenesis by regulating angiogenesis, immunity, and metastasis [7–9]. This provides a growing demand for simple, efficient, and affordable techniques to isolate exosomes. At the moment, the isolation of exosomes with reliable quality and substantial concentration is still a major challenge.

To date, several approaches for exosome isolation have been developed. These include differential ultracentrifugation, size-based ultrafiltration, and microfluidics-based platforms [10,11]. Besides, there are specific exosome precipitation methods, such as immunoaffinity capture-based techniques or lectin-based purification, and non-specific precipitation by PEG, alginic acid and hydrophobic binding [12–14]. According to the International Society for Extracellular Vesicles (ISEV) guidelines, isolated EVs are considered to be exosomes if they are in size range 30–200 nm, have a typical spherical form, contain a bilayer membrane, and are enriched with exosomal markers [15]. Here we compared EVs extracted from human plasma by five different methods: differential ultracentrifugation, sequential ultracentrifugation in a sucrose cushion, sedimentation of EV lectin aggregates, immunoprecipitation of exosomes and SubX™ technology. This new approach to the isolation of vesicles is based on using the proprietary bi-functional compound (SubX™) that can bind clusters of phospholipids on the vesicular surface and thus oligomerize vesicles directly in biological liquids [16]. The subsequent centrifugation precipitates [oligoEVs-SubX™] complex. The pelleted EVs dissociate back to monomers in the reconstruction buffer in a ready-to-use form for downstream applications. The vesicles isolated by the five techniques were characterized based upon size, quantity, CD63, CD81 and Calnexin protein expression, and total protein quality using several complementary methods: nanoparticle tracking analysis, dynamic light scattering, atomic force microscopy, cryo-electron microscopy, and flow cytometry.

## Materials and methods

The following reagents were used in the study: FITC-conjugated antibodies to CD63 and APC-conjugated anti-CD81 antibody (Beckman Coulter, USA); exosome isolation kits and kits for detection of the surface exosomal markers by flow cytometry (Lonza, Estonia), SubX™-Exo-DNA Plasma isolation kit (Capital Biosciences, USA), bovine pulmonary surfactant derived liposomes (Surfactant-HL, BIOSURF Ltd.), Bradford reagent and Clarity Western ECL Blotting Substrate (BioRad, USA); rabbit polyclonal antibodies to Calnexin (Abcam, ab22595). All other reagents used in the study were obtained from Sigma-Aldrich (USA).

## Plasma sampling and isolation of extracellular vesicles

The plasma samples from healthy donors were obtained from Blood Transfusion Unit of N.N. Petrov National Medical Research Center of Oncology; in accordance with the legislation of the Russian Federation, written informed consent was obtained from each donor. Clinical data were depersonalized. The study protocol (AAAA-A18-118012390156-5) was approved by the Ethics Committee of N.N. Petrov National Medical Research Center of Oncology. Blood samples were collected using EDTA-coated vacutainers. Plasma samples were isolated from peripheral blood using centrifugation at 3,000 rpm at 4 °C for 20 minutes and stored at -80 °C. Samples were thawed on ice immediately before analysis.

Extracellular vesicles were isolated from equal volumes (1 mL) of initial plasma after the preliminary removal of cellular debris and large vesicles by centrifugation (2,000 $g$ for 30 min, and then 16,000 g for 30 min). Within the framework of the study, EVs were isolated by the following methods:

1. Sequential ultracentrifugation (UC) was performed using the method described earlier [17,18]; it included the UC of plasma (diluted 1:5 with PBS) on a Beckman Coulter centrifuge (SW 55Ti rotor) at 110,000 g for 2 h. After centrifugation, the supernatant was removed, and the pellet was re-suspended in 1 mL of phosphate-buffered saline (PBS) for at least 1 h at 4°C; then the volume was adjusted to 5 mL and re-centrifuged at 110,000 $g$ for 2 h. The resulting pellet was dissolved in 100 μL of PBS.

2. Sequential UC in a sucrose cushion (UC + Suc) was carried out as described earlier [17]. Briefly, EVs were concentrated by the UC of plasma (diluted 1:5 with PBS) at 110,000 g (SW 55Ti rotor) for 2 h. After centrifugation, the supernatant was removed and the pellet was re-suspended in 4 mL of PBS for at least 1 h at 4°C. The resulting suspension was applied onto 1.5 mL of the sucrose cushion (30% sucrose/Tris/$D_2O$, pH 7.4) and re-centrifuged at 110,000 g for 2 h. After centrifugation, 1 mL of the cushion containing vesicles was gently taken by a syringe from the bottom of the tube. EVs were then concentrated and transferred to PBS by using Amicon-Ultra concentrators, 100 kDa (Millipore, USA). The final volume of the EV suspension was 100 μL.

3. Exosome isolation based on sedimentation of their lectin aggregates was carried out by the procedure described earlier [19,20]. Concanavalin A (Con-A) was used as a lectin binding sugar residue on the EV surface. This isolation method included the following steps: a) overnight incubation of the supernatant, in the presence of Con-A (1.0 μg/mL) at 4°C under constant stirring to form the EV aggregates; b) precipitation of EV aggregates by centrifugation at 15,000 $g$ for 30 min; c) washing the EV precipitate with PBS and repeating the previous step; d) EV disaggregation in an excess of monosaccharides (40% glucose in PBS) for 4 h; e) ultrafiltration (Amicon-Ultra, 100 kDa, Millipore) and washing with PBS for purification of isolated EVs from Con-A and concentration of the preparation. The final volume of the isolated EV preparation was 100 μL.

4. Isolation of exosomal EVs by immunoprecipitation from plasma (1 mL) was carried out using the exosome isolation kit (Lonza, Estonia), which contains antibodies to the surface exosomal marker CD9, in accordance with the manufacturer's recommendations. Briefly, 10 μl of pre-coupled beads were incubated overnight at 4°C in rotator with 1 mL of pre-cleared plasma. After exosome binding, the beads were washed twice with 1 mL of PBS and centrifugated at 5,000 g for 10 min. Exosome elution from beads was performed by incubation with 10 μl of Exosome Elution Buffer for 5 min. Then exosomes were eluted in 100 μL of PBS by centrifugation at 5,000 g for 10 min.

5. Isolation of EVs by the SubX reagent included the following steps: a) centrifugation of plasma (1 mL diluted 1:20 with PBS) at 10,000 g for 1 h.; b) 30 min incubation of the supernatant, in the presence of 100 μL SubX reagent (1/10 of the plasma initial volume) at room temperature to form the EV aggregates; c) precipitation of EV aggregates by centrifugation at 10,000 $g$ for 30 min; d) washing the EV precipitate with PBS and repeating the previous step; e) EV disaggregation in 300 mM NaCl for 1 h; f) re-centrifugation of the obtained suspension at 10,000 $g$ for 30 min to remove possible contaminations of large aggregates; at this step, the pellet is thrown away, and the supernatant is used for the following step; g) ultrafiltration (Amicon-Ultra, 100 kDa, Millipore) and washing with PBS for the purification of isolated EVs from SubX reagent and the concentration of the preparation. The final volume of the isolated EV preparation was 100 μL.

Thus, five EV preparations (100 μL) isolated by different methods from identical samples of plasma (1 mL) were obtained. The EV samples were aliquoted, rapidly frozen in liquid nitrogen and stored at -80˚C until analysis.

## Methods for analysis of isolated extracellular vesicles

The EV size distribution was evaluated by the method of dynamic light scattering (DLS) using the particle size and zeta potential analyzer Photocor Compact-Z (Photocor Ltd., Russia) or Zetasizer Nano-ZS (Malvern Instruments, UK) according to the manufacturer's recommendations. All measurements were carried out at +25˚C. For each sample the particle size distribution curves were plotted by results of three measurements.

The EV size and concentration were determined by nanoparticle tracking analysis (NTA) using the NTA NanoSight® LM10 (Malvern Instruments, UK) analyzer, equipped with a blue laser (45 mW at 488 nm) and a C11440-5B camera (Hamamatsu Photonics K.K., Japan). Recording and data analysis were performed with the use of the NTA software 2.3. The following parameters were evaluated during the analysis of recording monitored for 60s: the average hydrodynamic diameter, the mode of distribution, the standard deviation, and the concentration of vesicles in the suspension.

The detection of EVs was carried out by the atomic force microscopy (AFM). Briefly, the EV solution in PBS was diluted 50 times in deionized water, and 10 μL were put on freshly cleaved mica. After 1min incubation at room temperature the mica surface was thrice washed by water to remove salt. The sample topography measurements were performed in semi-contact mode on atomic force microscope "NT-MDT-Smena B". A probe NSG03 was used (NT-MDT, Russia). The images were analyzed using "Gwyddion" software [21].

Morphology of the isolated EVs was assessed using cryo-electron microscopy (cryo-EM) as described [22]. The initial volume of plasma for the study of EVs using the cryo-EM was 10 mL. The study was carried out on a Titan Krios 60–300 TEM/STEM (FEI, USA) transmission electron microscope equipped with a highly sensitive direct electron detector (DED) Falcon II (FEI) and a spherical aberration corrector (CEOS, Germany). Electron microscopy copper grids coated with a thin layer of amorphous carbon were treated in a Pelco easiGlow unit to produce a hydrophilic surface. After that, 3 μL of an analyzed sample was then applied onto the grids, and the sample was immediately frozen by a VitrobotMarkIV (FEI) unit in liquid ethane cooled to liquid nitrogen temperature (-196˚C). The samples were thus fixed in a thin layer of amorphous ice, which allowed the investigation of EVs in their native state. In order to minimize radiation damage, the data set was performed using the EPU (FEI) software operated at a low dose mode.

Analysis of the exosomal markers (tetraspanins CD63 or CD81) on the surface of the isolated EVs was carried out using Exo-FACS ready-to-use kit for plasma exosome analysis

(Lonza, Estonia) according to the manufacturer's recommendations. Bead-coupled EVs were assayed using FITC-conjugated anti-CD63 antibody (Beckman Coulter, USA) and APC-conjugated anti-CD81 antibody (Beckman Coulter, USA). Analysis was performed with CytoFlex instrument (Beckman Coulter, USA).

The presence of the negative exosomal marker Calnexin in the samples of isolated vesicles was determined by western blotting. The 5 samples of isolated EVs from the same amount of plasma were incubated at 4˚C for 30 minutes with 20 μL of lysis buffer (7M urea, 2M thiourea, 4% CHAPS, 5 mM PMSF, 1% DTT). The protein samples were diluted in Laemmli buffer (BioRad, USA), subjected to 10% SDS-PAGE containing 0.1% SDS, and transferred to the PVDF membrane (Thermo Scientific) using the Trans-Blot Turbo Transfer System (BioRad, USA). Rabbit polyclonal antibodies to Calnexin (Abcam, ab22595) at dilution of 1:200 were used as primary antibodies. Horseradish peroxidase-conjugated goat antibodies against mouse immunoglobulins from Sigma were used as secondary antibodies at dilutions of 1:10,000. Chemiluminescent detection of the protein bands was performed with Clarity Western ECL Blotting Substrate (Bio-Rad, USA) and Thermo Scientific CL-XPosure Films (Thermo Fisher Scientific, USA). U-87 MG ell lysate (in the same amount of total protein) was used as a positive immunodetection control.

The total protein content in EV samples was determined using a colorimetric method (Bradford) after lysis of vesicles in a buffer containing urea (7M urea, 2M thiourea, 4% CHAPS, 1% DTT). Analysis was performed with BioRad SmartSpec Plus instrument (BioRad, USA).

## Results

### SubX™ technology validation

An approach to isolating EVs based on the phenomenon of binding and aggregation of these particles via clusters of outer membrane phosphate groups in the presence of SubX™ reagent (Capital Biosciences, USA) has been put forward in the present study. It is assumed that, since vesicular membrane phospholipids contain multiple phosphate groups, this can lead to the formation of [SubX+EVs] aggregates. We first tested this feature on a model system. Thus, to confirm phosphate group clusters mediated mechanism of lipid vesicle aggregation via SubX™ anchoring, we performed a model experiment, which includes SubX™ addition to phospholipid-based and phosphatidylcholine-based vesicles with blocked phosphate groups. As phospholipid-based vesicles we employed commercially available liposomes manufactured from natural bovine lung extract (Surfactant-HL, BIOSURF Ltd.). As phospholipid particles with blocked phosphate groups we employed an in-house prepared phosphatidylcholine emulsion of exosomal size. Vesicles were analyzed by DLS to determine the particle size distribution in the total light scattering of the analyzed liquid sample. As seen from Fig 1, the liposomes derived from the emulsion of bovine lung alveolar liquid with phospholipide content of 80% have a bimodal particle size distribution with the maximum at 100 nm and 500 nm (Fig 1A, red line). SubX™ efficiently binds and oligomerizes phospholipid enriched vesicles into aggregates with the mean size of ~1000 nm (Fig 1A, blue line). In contrast, phosphatidylcholine-derived vesicles with blocked phoshate residues (Fig 1B, red line) do not interact with SubX™, and their size distribution pattern remains the same after the addition of SubX™ (Fig 1B, blue line). These data strongly confirm the suggestion that SubX™ captures phosholipid-containing vesicles through phosphate moietie clusters displayed on outer membrane.

Next, we tested the assumption that clusters of phospholipids on the surface of plasma vesicles can also be bound by the SubX™ reagent, leading to the aggregation of the EVs. To do this, we tried the following procedure: the blood plasma (1mL) diluted at 1:20 by PBS was

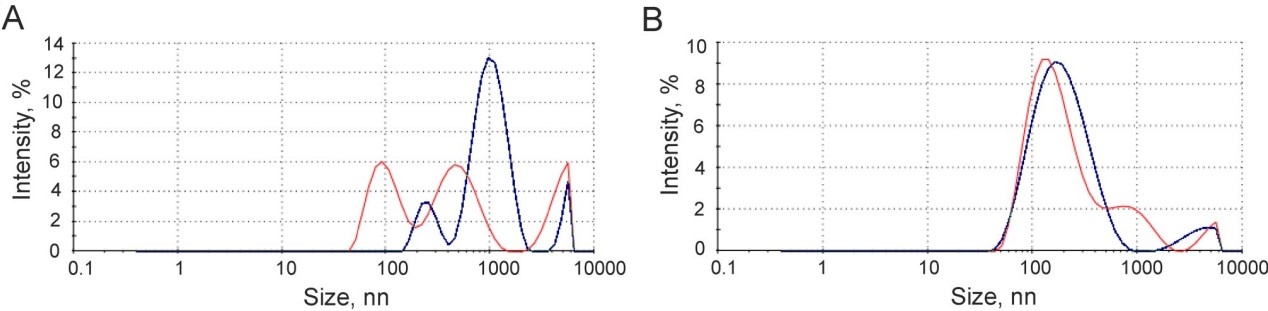

**Fig 1. Effect of SubX™ on phospholipid-based vesicles (A) and phosphatidylcholine-based vesicles (B).** Vesicles were analyzed by dynamic light scattering (DLS) to determine the particle size distribution in the presence (blue line)/absence (red line) of the SubX™ reagent.

centrifuged at 10,000 g for 1 h to remove large particles and cellular debris; then 100 μL of SubX™ reagent (1/10 of the plasma initial volume) was added to the supernatant and incubated 30 min at room temperature; after that, the sample was centrifuged at 10,000 *g* for 30 min, the supernatant was removed, and the pellet was re-suspended in PBS; then the sample was centrifuged at 10,000*g* for 30 min again, and the final pellet was re-suspended in a 300 mM solution of NaCl. At each stage of this procedure, an aliquot of the sample was analyzed using DLS. The results are shown in Fig 2.

In the supernatant obtained after the first centrifugation of diluted plasma, only a small number of particles with exosomal size were observed. Most likely, this is due to the masking of exosome-like vesicles by a large number of proteins present in the plasma and identified on the histogram as a peak with a particle size of about 10 nm (Fig 2A). Most of these proteins remain in the untargeted supernatant after the incubating of the sample with the SubX™ reagent and the first centrifugation at 10,000 g (Fig 2B). At the same time, particles with sizes of about 300 nm were observed in the re-suspended pellet obtained after the incubation with the SubX™ reagent (Fig 2C). In addition, the peak corresponding to particles with sizes of about 10 μm was significantly larger than that in all analyzed aliquots (Fig 2C). These data suggest that the addition of SubX™ to diluted plasma resulted in the appearance of the peaks corresponding to molecular aggregates substantially larger than exosomes. These aggregates could be sedimented by centrifugation at low speed (10,000 *g*). The addition of 300 mM NaCl to the final pellet resulted in the destruction of the bonds between the molecules of SubX™ and

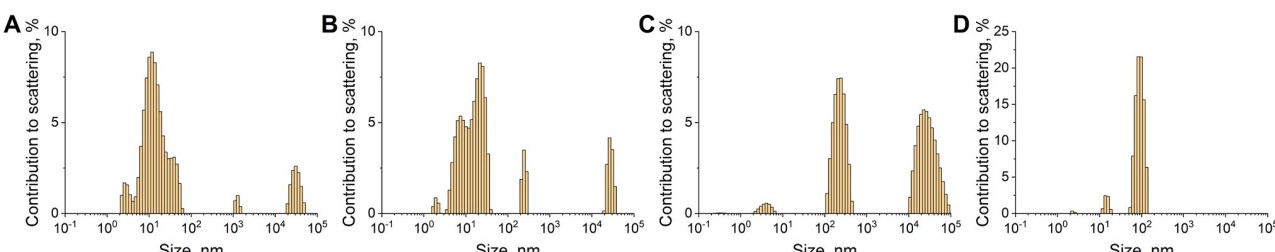

**Fig 2. Aggregation of plasma exosome-like vesicles by a SubX™ reagent.** Particle size distribution in: the initial plasma pre-cleaned from cell debris (A); the supernatant (B) and resuspended pellet (C) obtained after incubation of precleaned plasma with SubX™SubX and the first centrifugation at 10,000 g; the final pellet after the second round of centrifugation at 10,000 g and resuspension of the precipitate in a 300 mM solution of NaCl (D). Data obtained by dynamic light scattering (DLS). The X-axis is the hydrodynamic diameter of the particles in nm, the Y-axis is the contribution to the scattering in %.

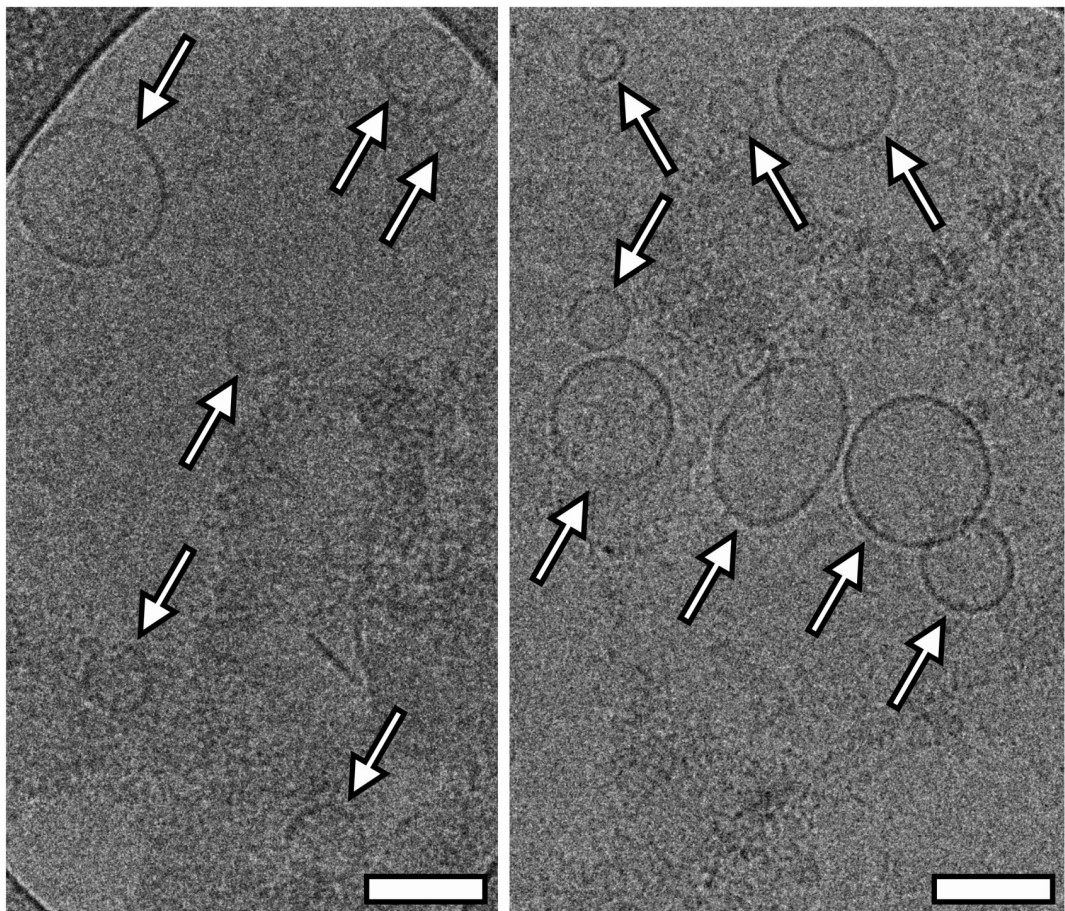

**Fig 3. Cryo-EM images of extracellular vesicles isolated from plasma by SubX™ reagent.** The white arrows point to the vesicles with a double membrane. Scale bars are 100 nm.

phospholipid clusters on the surface of EVs and the release of single vesicles from the aggregates. Disaggregated exosome-like vesicles have been detected by DLS as a peak with the size of about 90 nm (Fig 2D). These results clearly indicate the presence of the sites for the binding of SubX™ on the outer membrane of the vesicles, which confirms the possibility of EV separation from the plasma using SubX™ technology. A small peak corresponding to a particle size of about 10 nm indicates insignificant contamination of the EV sample with supposedly phosphorylated molecules (Fig 2D).

In order to confirm the vesicular nature of particles isolated with SubX™, these samples were analyzed by cryo-EM. As seen in Fig 3, isolated particles are indeed formed by a characteristic lipid bilayer, which has an average thickness of ∼10 nm, and have a round-shaped vesicular morphology. The visualized particles with a lipid bilayer generally varied in size from 50 to 220 nm (Fig 3).

## Comparison of SubX technology with conventional methods (physical properties)

The isolation of the circulating nano-vesicles (exosomes) from plasma is usually performed by one of the following methods: ultracentrifugation (UC), sequential UC in a "sucrose cushion",

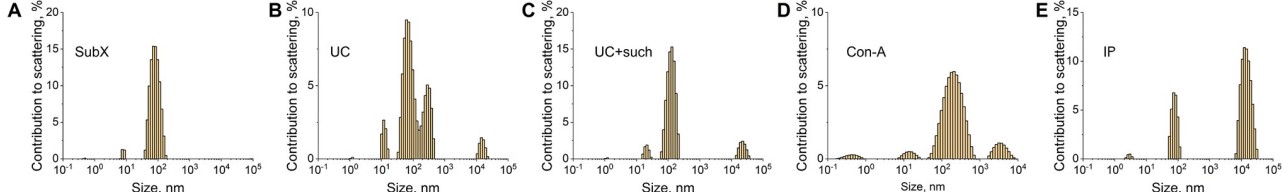

**Fig 4. Evaluation of the distribution of extracellular vesicles (EVs) in size by the method of dynamic light scattering (DLS).** Designation of the isolation methods: SubX™ technology (A); Ultracentrifugation–UC (B); ultracentrifugation in the cushion of 30% sucrose—UC + Suc (C), precipitation of lectin aggregates—Con-A (D); immunoprecipitation–IP (E). The X-axis is the hydrodynamic diameter of the particles in nm, the Y-axis is the contribution to the scattering in %.

agglutination by natural or synthetic polymers, capture by immune-beads with antibodies against known exosomal markers. Further in this study, we isolated EVs from equal samples of human plasma (1 mL) using SubX™ technology and the four methods mentioned above in parallel. Five samples of isolated vesicles were analyzed by DLS, nanoparticle tracking analysis (NTA) and atomic force microscopy (AFM). Presence of the exosomal surface markers CD63 and CD81 in membrane of isolated vesicles was evaluated by flow cytometry.

DLS was used to estimate the averaged hydrodynamic radius of isolated vesicles (Fig 4). Thus, a peak of 100 nm corresponding to the size of exosome-like vesicles was observed in all samples of EVs isolated by different methods. However, only the SubX™ protocol resulted in the isolation of predominantly 100 nm vesicles with minor contamination by plasma proteins (about 5–20 nm) (Fig 4A). Different particles sized from 50 nm to 200 nm contaminated by proteins (about 5–20 nm) and large (>10 µm) particles were isolated by UC (Fig 4B). UC with a "sucrose cushion" yielded a population of vesicles with a predominant fraction sized 100 nm (Fig 4C); however, small (<1 nm) and large (>10 µm) particles are still present. Thus, the use of "sucrose cushion" during UC increases the purity of the exosome-like fraction of isolated vesicles. The method of EV isolation involving agglutination by lectins, followed by sedimentation and dis-agglutination with monosaccharides, resulted in the isolation of particles of different sizes, including 100 nm. However, smaller particles and large aggregates, probably stemmed from non-complete disaggregation of EVs, were observed (Fig 4D). Isolation with immune-bead bearing antibodies against exosomal marker CD9 results in the isolation of an almost pure population of exosomes. A large peak in Fig 4E corresponding to the size of several micrometers reflected the presence of immune-beads in the sample.

At the next stage, NTA was applied to confirm measurement of EV size and to estimate their concentration (Fig 5). The mode size of vesicles isolated by all methods was in a range from 63 to 99 nm, which corresponded well with the size of exosomes. However, concentrations of vesicles isolated by different methods were quite different. Ultracentrifugation allowed to obtain the highest concentration of vesicles–$(7.8 \pm 0.7) \times 10^{11}$ particles/mL (Fig 5B). The use of "sucrose cushion" considerably reduced the yield of vesicles–$(2.3 \pm 0.4) \times 10^{11}$ particles/mL (Fig 5C). An even lower concentration of vesicles was obtained by other methods: immune capturing $(1.1 \pm 0.3) \times 10^{11}$ particles/mL (Fig 5E), SubX™ $(0.7 \pm 0.2) \times 10^{11}$ particles/mL (Fig 5A) and agglutination by lectins $(0.4 \pm 0.1) \times 10^{11}$ particles/mL (Fig 5D).

Surface topology of plasma nanovesicles was estimated by AFM. In five samples of vesicles isolated by different methods, we have observed individual particles of a spherical shape that corresponds to vesicular topology (Fig 6A and 6C–6F). The population of vesicles isolated by SubX™ was analyzed in more details. Individual vesicles examined under AFM show characteristic cup-shaped morphology, appearing as flattened spheres with diameters ranging from 30 to 70 nm (Fig 6B). The cup-shaped morphology is most likely originated from the sample

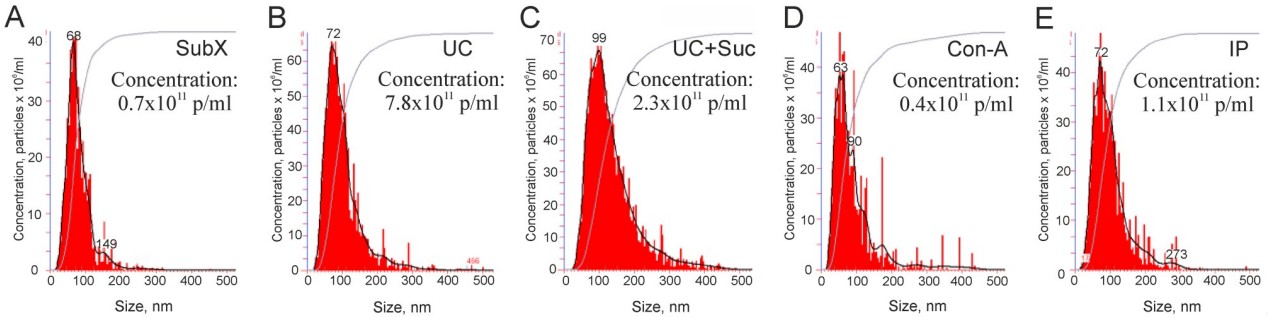

**Fig 5. Nanoparticle tracking analysis (NTA): Size distribution (nm) and particle concentration (x10$^{11}$ particles/mL) in preparations of extracellular vesicles.** Designation of the isolation methods: SubX™ technology (A); Ultracentrifugation–UC (B); ultracentrifugation in the cushion of 30% sucrose—UC + Suc (C), precipitation of lectin aggregates—Con-A (D); immunoprecipitation–IP (E).

preparation process of conventional AFM microscopy, during which the vesicles are extremely dehydrated.

## Comparison of SubX™ technology with conventional methods (molecular content)

As suggested previously by the international society of EVs [15], we have analyzed a set of proteins, which should either be present in or excluded from the exosome population. In order to confirm the exosomal nature of isolated vesicles, they were non-specifically absorbed on the flow cytometry beads (4 μm) followed by incubation with antibodies to exosomal markers (CD63 or CD81). The presence of exosomal markers on the surface of vesicles was estimated in parallel in five samples of vesicles isolated by different methods using the same exosomal

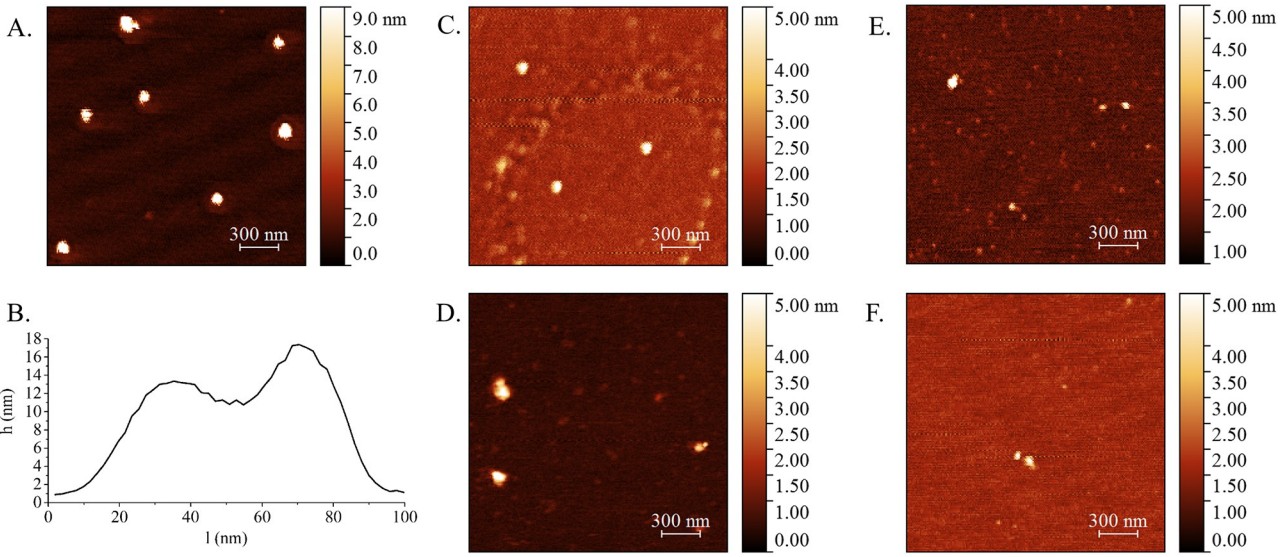

**Fig 6.** The surface topography of vesicles isolated by SubX™ technology (A), ultracentrifugation (C), ultracentrifugation in 30% sucrose (D), precipitation of lectin aggregates (E); immunoprecipitation (F). The characteristic "cup-shape" particle profile, presented on (B), "h"–vesicular height (nm) and "l"–diameter (nm). The scale bars are 300 nm. On the right of (A)-(F) is the pseudo color ruler indicating the particles' height (nm).

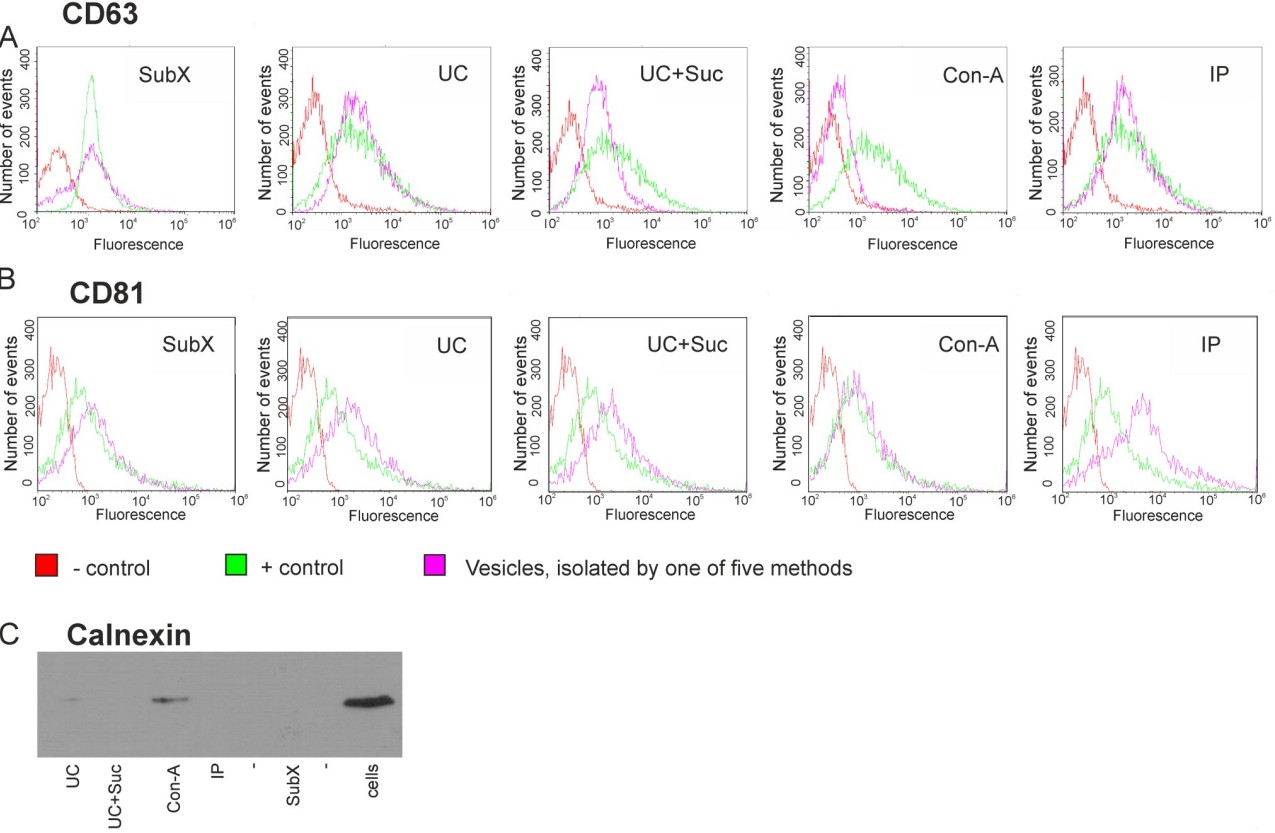

**Fig 7. Analysis for positive and negative exosome markers in the samples of vesicles isolated by 5 methods.** Flow cytometry analysis of isolated vesicles for the surface expression of CD63 (A) and CD81 (B) tetraspanins classically used as exosome markers. Immunobeads blocked with BSA and stained with anti-CD63 or CD81 antibodies were used as negative control (–control). The exosomal standard included in the exosome cytometric assay kit (Lonza) was used as a positive control (+ control). Western blot analysis for exosome negative marker, calnexin, in isolated samples of vesicles and U-87 MG cell lysate sample (C). Designation of the isolation methods: SubX™ technology; Ultracentrifugation–UC; ultracentrifugation in the cushion of 30% sucrose—UC + Suc, precipitation of lectin aggregates—Con-A; immunoprecipitation–IP.

standard as positive control and beads non-incubated with exosomes as a negative control. The presence of CD63 and CD81-positive vesicles was confirmed by flow cytometry in all samples including the sample of EVs isolated by SubX™ (Fig 7A and 7B).

Proteins associated with compartments other than plasma membrane or endosomes should not be detectable within exosomes [17]. In fact, we were able to show that Calnexin, a protein associated with the endoplasmic reticulum, was found in whole U-87 MG cells, to a much lesser extent also in the sample of vesicles isolated with Con-A, but was excluded from all samples of vesicles isolated by other methods (Fig 7C).

Finally, total protein content of the vesicles isolated from the same amount of plasma by different approaches was tested (Table 1). Protein concentration seems to reflect the concentration of vesicles in the samples determined by NTA: the highest concentrations of proteins were detected in samples isolated by UC (0.45±0.08), with the highest concentration of vesicles being $7.8 \times 10^{11}$, and UC with "sucrose cushion" (0.18±0.03), with the concentration of vesicles being $2.3 \times 10^{11}$. Isolation of vesicles by lectin and SubX™ yielded to low concentrations of both vesicles and proteins.

**Table 1. Total protein content in vesicles isolated by methods: Ultracentrifugation—UC; ultracentrifugation in the cushion of 30% sucrose—UC + Suc; precipitation of lectin aggregates—Con-A; immunoprecipitation—IP; SubX™ technology.**

|  | Total protein (mg/ml) (Bradford assay) |
|---|---|
| UC | 0.45±0.08 |
| UC+Suc | 0.18±0.03 |
| Con-A | 0.10±0.01 |
| IP | 0.15±0.05 |
| SubX™ | 0.12±0.07 |

## Discussion

As cell-derived extracellular vesicles, exosomes play a significant role in intercellular communication by serving as a carrier for the transfer of proteins, lipids, and RNA between cells [3]. Exosomal components have been found to be related to certain diseases and treatment responses, including cancer, neurodegenerative, and cardiovascular diseases. Therefore, they are considered to be crucial for the discovery of biomarkers for clinical diagnostics [23–25]. In addition, several studies have reported that EVs can be employed as a drug delivery system for targeted therapies with drugs and biomolecules [26,27]. The isolation of EVs or pure exosomes is critical for their subsequent analysis and applications in biomedical sciences. Various techniques have been adopted to facilitate the purification of exosomes [24,27]. Nevertheless, isolation of exosomes is still a state-of-art technique due to their vesicular nature–heterogeneity of particle size and shape pattern, variable membrane composition, and bio-liquid specificity. Here we compared the EVs isolated from equal volumes of human plasma by five different methods: differential ultracentrifugation, sequential ultracentrifugation in a sucrose cushion, sedimentation of EV lectin aggregates, immunoprecipitation of exosomes using commercial kit, and by SubX™ technology.

Ultracentrifugation-based EV isolation is considered to be the gold standard and is one of the most commonly used and reported techniques for exosome isolation [28,29]. Here we used two types of preparative ultracentrifugation–differential ultracentrifugation and sequential ultracentrifugation in a sucrose cushion. Based on a comparative analysis of the five differently isolated samples from plasma, we can conclude that the highest yield of exosomes was achieved by means of ultracentrifugation. However, exosomes obtained by this method are characterized by the highest level of contaminations with non-vesicular particles. Combination of the UC with the sucrose cushion reduces the level of non-vesicular contamination, but the number of isolated vesicles drastically decreases. Similar data were shown in a number of previous studies that analyzed the quality and quantity of vesicles isolated from various biological fluids by these methods [30–32].

Most exosomes contain proteins that are common among all exosomes regardless of the types of cells which secrete them [33,34]. These characteristic proteins therefore may serve as convenient biomarkers for the isolation and quantification of exosomes. However, several earlier studies demonstrated variability of vesicular populations secreted even by the same cells in terms of membrane protein content and biological properties [35–37]. Different approaches to separate vesicles on the basis of their physical properties (by differential centrifugation [38] or asymmetric flow field-flow fractionation [39]) also revealed distinct subsets of extracellular vesicles. Moreover, proteomic analysis [40] as well as deep RNA sequencing [41] have confirmed the existence of heterogeneous populations of extracellular vesicles of exosomal nature. Here we used plasma exosome purification kit (Lonza, Estonia) for immunoprecipitation of

exosomes. Based on the results, we can state that the vesicle preparation obtained in this way is the most enriched with CD63 or CD81-positive vesicles; however, the number of particles released from immune-beads is extremely low compared to other approaches. It may be correct to say that immunoprecipitation allows to isolate specific populations of exosomes enriched with a certain surface marker, but not complete fraction of 100 nm exosomal vesicles. Thus, such extreme specificity of the isolation procedure is not always desirable.

Recently, a number of non-immunological precipitation procedures have been launched to isolate and study vesicles for various purposes [24,27]. Compared to ultracentrifugation, these procedures are less time consuming, less technique sensitive, and do not require special equipment. Most of these isolation methods are based on the composition of microvesicle membranes. For example, the outer surface of EVs and exosomes is rich in saccharide chains, such as mannose, polylactosamine, alpha-2,6 sialic acid, and N-linked glycans [42]. For our comparative analysis we used a preparative technique for isolation of EVs based on their ability to aggregate in the presence of lectins. The method for lectin-based isolation of exosomes from different biological fluids was tested earlier for exosome-based protein or miRNA biomarker researches [12,19,20]. In the comparative analysis of the methods for isolating vesicles that we conducted in this study, the isolation of EVs by precipitation of lectin aggregates proves significantly inferior to all other methods in terms of purity and quantity of the final sample. These data are consistent with the observations we made earlier when exosomes were isolated from the culture medium [32]. The advantages of the Con-A method are technical simplicity, the ability to handle large initial volumes of biological fluid, and a relatively low cost.

SubX™ is the alternative innovative technology based on affinity capture of membrane phosphate moieties of two neighbor vesicles via bifunctional SubX™ molecule. Since vesicular membrane phospholipids contain multiple phosphate groups, it results in an assembly of [SubX™+EVs] oligomers that precipitate from a complex bio-liquid by conventional bench-top centrifugation step. The pelleted EVs dissociate back to monomers in the reconstruction buffer in a ready-to-use form for downstream applications. The basic characteristics of SubX™ technology and four other methods are compared in Table 2.

The results of NTA or DLS demonstrate the presence of particles with a characteristic exosome size in the final sample isolated by SubX™. Comparison Sub-X technology with other methods indicates its relatively low efficacy (yield), yet high specificity (purity). In terms of other characteristics, SubX™ technology is not labor consuming, is well scalable and can be easily standardized. Cytometric detection of the CD63 and CD81 markers on the surface of the particles confirms the exosomal nature of the vesicles isolated from human plasma by SubX™. Cryo-electronic and atomic force microscopy analyses demonstrate that SubX™ technology allows obtaining homogeneous exosomal-size particles in the final preparation. Cryo-EM does

**Table 2. Summary of different exosomes isolation methods.**

|  | Yield* | Purity** | Time, hours | Scalability | Standardization |
|---|---|---|---|---|---|
| UC | 7.8 ± 0.7 | 55,3 | 7–8 | Hardly | Hardly |
| UC+Suc | 2.3 ±0.4 | 79,1 | 8–9 | Very hardly | Very hardly |
| Con-A | 0.4±0.1 | 42,6 | 18–20 (ON) | Intermediate | Hardly |
| IP | 1.1±0.3 | 96,1 | 14–16 (ON) | Well | Well |
| SubX™ | 0.7±0.2 | 92,8 | 5–6 | Well | Well |

*The yield of exosome isolation reflecting efficacy of method is presented as results of NTA (particles concentration, x10^11 particles / mL);

** The purity of isolated exosomes is presented as results of DLS (% of ≈ 100 nm particles in the whole measured population of particles), the peak corresponding to immunobeads used for IP is excluded from calculation. ON is an overnight incubation.

not suffer from the effects of dehydration and fixation issues. Thus, Cryo-EM is considered the best method for visualizing extracellular vesicles and proteins without dehydration artifacts [22,43]. In this study, we capture images of exosomes isolated by SubX™, with membrane structures and lumens.

## Conclusion

Summarizing all the data, we can assume that SubX™ allows obtaining relatively pure populations of exosomes, while the concentration of isolated vesicles is rather low. Undoubtedly, this method is suitable for the isolation of EVs from human blood plasma. The method is rather simple and does not require complex, expensive equipment. Its scope is limited to a relatively low yield of the target product; however, a more comprehensive analysis of SubX™-isolated EVs may help to further our insight into the complex composition of plasma exosomes.

## Supporting information

**S1 Fig.**
(TIF)

## Acknowledgments

The authors acknowledge the support and the use of resources of the Resource Center for Probe and Electron Microscopy at the NRC "Kurchatov Institute".

## Author Contributions

**Conceptualization:** Vladimir Evtushenko, Anastasia Malek.

**Formal analysis:** Anastasia Malek.

**Funding acquisition:** Tatiana Shtam.

**Investigation:** Tatiana Shtam, Roman Samsonov, Roman Kamyshinsky, Vladimir Burdakov.

**Methodology:** Tatiana Shtam, Vladimir Evtushenko, Andrey Konevega.

**Resources:** Andrey Konevega, Anastasia Malek.

**Validation:** Roman Samsonov, Yana Zabrodskaya, Lidia Zabegina, Nikolay Verlov, Luiza Garaeva, Maria Slyusarenko, Nadezhda Nikiforova.

**Visualization:** Yana Zabrodskaya, Roman Kamyshinsky.

**Writing – original draft:** Tatiana Shtam, Vladimir Evtushenko.

**Writing – review & editing:** Tatiana Shtam, Vladimir Evtushenko, Yana Zabrodskaya, Anastasia Malek.

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
