## [Decision Letter · Decision Letter 0]

4 Aug 2020

PONE-D-20-14332

Evaluation of immune and chemical precipitation methods for plasma exosome isolation

PLOS ONE

Dear Dr. Shtam,

Thank you for submitting your manuscript to PLOS ONE. After careful consideration, we feel that it has merit but does not fully meet PLOS ONE’s publication criteria as it currently stands. Therefore, we invite you to submit a revised version of the manuscript that addresses the points raised during the review process.

We look forward to receiving your revised manuscript.

Kind regards,

Girijesh Kumar Patel, PhD

Academic Editor

PLOS ONE

Journal Requirements:

2.In your Data Availability statement, you have not specified where the minimal data set underlying the results described in your manuscript can be found. PLOS defines a study's minimal data set as the underlying data used to reach the conclusions drawn in the manuscript and any additional data required to replicate the reported study findings in their entirety. All PLOS journals require that the minimal data set be made fully available. For more information about our data policy, please see http://journals.plos.org/plosone/s/data-availability.

Reviewers' comments:

Reviewer's Responses to Questions

**Comments to the Author**

1. Is the manuscript technically sound, and do the data support the conclusions?

Reviewer #1: Yes

Reviewer #2: Yes

2. Has the statistical analysis been performed appropriately and rigorously? 

Reviewer #1: Yes

Reviewer #2: Yes

3. Have the authors made all data underlying the findings in their manuscript fully available?

Reviewer #1: Yes

Reviewer #2: Yes

4. Is the manuscript presented in an intelligible fashion and written in standard English?

Reviewer #1: Yes

Reviewer #2: Yes

5. Review Comments to the Author

Reviewer #1: In this manuscript by Tatiana Shtam et al., the authors evaluated several different methods for isolation of plasma derived exosomes.

The experiments were expertly performed and were appropriately controlled. The data presentation is good. The experimental design is appropriate and the data are convincing. The manuscript is clearly written.

However, there are some major concerns:

- To estimate the purity of isolated exosome fractions it would be necessary to provide the analysis of non-EV-associated molecules, such as Apolipoproteins, to see how much contaminants are co-isolated in each method.

In lines 272-274 the authors write e.g. "Without contamination by plasma proteins", but just determined this on the basis of size.

The MISEV 2018 guidelines provide a table with categorized proteins that should be used to analyze the nature and purity of EVs (https://www.tandfonline.com/doi/full/10.1080/20013078.2018.1535750, table 3). This was not fully considered by the authors, since only CD63 was analyzed (e.g. line 431-433: Analysis of CD63 alone is not sufficient to confirm that these particles are exosomes).

- The authors did not use a uniform nomenclature for the vesicles they isolated. Sometimes the term “EVs” was used, then “exosomes” or “exosome-like vesicles”.

- The English could be improved, sometimes the authors use wrong words, e.g. in line 421 “adventures” instead of “advantages”.

Reviewer #2: In this manuscript, authors present a comparison of five protocols for exosome isolation from human

plasma: two chemical precipitation methods (lectin-based purification and SubX TM

technology), immunoaffinity precipitation, vs. basic ultracentrifugation-based exosome

isolation method in two modifications.

The over all manuscript is well written and informative. The authors should proofread manuscript. Discussion section can be improved further by removing redundancy.

6. PLOS authors have the option to publish the peer review history of their article (what does this mean?). If published, this will include your full peer review and any attached files.

Reviewer #1: No

Reviewer #2: No

---

## [Author Response · Author response to Decision Letter 0]

2 Oct 2020

We thank the Editor and the Reviewers for the interest in our work and constructive comments. We tried to satisfy these remarks, if it was possible. All major modifications are highlighted in red inside of the manuscript. Besides, Fig.2-7 were modified, additional panels were added to Fig.7., Table2 was included, and additional refs (4 and 14) were added in Reference Section.

The detailed replies to Reviewer’s comment are below.

Reviewer #1: 

In this manuscript by Tatiana Shtam et al., the authors evaluated several different methods for isolation of plasma derived exosomes. The experiments were expertly performed and were appropriately controlled. The data presentation is good. The experimental design is appropriate and the data are convincing. The manuscript is clearly written.

However, there are some major concerns:

- To estimate the purity of isolated exosome fractions it would be necessary to provide the analysis of non-EV-associated molecules, such as Apolipoproteins, to see how much contaminants are co-isolated in each method.

>>Thank you very much for your fair comment. We have added Western blot analysis for the presence of Calnexin in isolated vesicle samples. Calnexin, a protein associated with the endoplasmic reticulum, is one of the negative markers of exosomes. The corresponding additions are made to Figure 7 and the revised manuscript to the Results and Materials & Methods sections.

In lines 272-274 the authors write e.g. "Without contamination by plasma proteins", but just determined this on the basis of size.

>>We absolutely agree that a too strong statement was made, not supported by additional methods. We have corrected the wording.

The MISEV 2018 guidelines provide a table with categorized proteins that should be used to analyze the nature and purity of EVs (https://www.tandfonline.com/doi/full/10.1080/20013078.2018.1535750, table 3). This was not fully considered by the authors, since only CD63 was analyzed (e.g. line 431-433: Analysis of CD63 alone is not sufficient to confirm that these particles are exosomes).

>>We agree with the comment of the reviewer about the insufficient determination of only one exosomal marker, and we followed their suggestions. We have added a flow cytometric analysis of the presence of tetraspanin CD81 on the surface of the five-way isolated vesicles. The corresponding additions are made to Figure 7 and the revised manuscript. 

The authors did not use a uniform nomenclature for the vesicles they isolated. Sometimes the term “EVs” was used, then “exosomes” or “exosome-like vesicles”.

>>Thank you for your comment, we have tried to correct the nomenclature variety.

The English could be improved, sometimes the authors use wrong words, e.g. in line 421 “adventures” instead of “advantages”.

>>English corrected throughout the text

Reviewer #2: 

In this manuscript, authors present a comparison of five protocols for exosome isolation from human plasma: two chemical precipitation methods (lectin-based purification and SubX TM technology), immunoaffinity precipitation, vs. basic ultracentrifugation-based exosome isolation method in two modifications.

The over all manuscript is well written and informative. The authors should proofread manuscript. Discussion section can be improved further by removing redundancy.

>>Thank you for your good evaluation of our study and helpful comments. The Discussion section is shortened in the revised manuscript. Language correction have been made across all the text of manuscript. 

Reviewer #3: 

The authors carry out a reasonably-designed comparison of 5 different methods of EV isolation from human plasma. Included is a stated ‘novel’ method based on a phosphate-driven aggregation based on the compound SubXTM. 

>>We thank the reviewer for the attention and suggestions that we believe will improve our manuscript.

Below is our point-by-point description of the revisions according to the reviewer’s comments.

We are never told the identity of the compound, and so are immediately put off as this should be an open description of science. We are never informed as to the commercial source of this reagent nor is there a specific reference to the origins of the methodology described on page 6. We find this to be disingenuous and to be grounds itself for rejection. 

>>In our early studies, we used the SubXTM Plasma cfDNA Isolation Kit to isolate extracellular DNA from plasma. The description of this reagent published on the company's website (https://www.capitalbiosciences.com/cfdna-isolation-reagents), namely the mention of the ability of SubXTM to bind phosphate residues, gave us the idea of using this reagent to isolate exosomes, the membrane of which is enriched in phospholipids. The presented study is devoted to the experimental verification of this idea. 

The technique for isolating exosomes using SubXTM reagent was tested and modified in the course of experiments. This technique was based on the procedure developed by us for the isolation of vesicles from biological fluids using the precipitation of lectin aggregates of exosomes. (Lectin‐induced agglutination method of urinary exosomes isolation followed by mi‐RNA analysis: Application for prostate cancer diagnostic. Prostate. 2016;76: 68–79. doi:10.1002/pros.23101; Shtam TA et al Aggregation by lectins as an approach for exosome isolation from biological fluids: Validation for proteomic studies. Cell tissue biol. 2017;11: 172–179; Samsonov et al., Shtam TA et al., Isolation of Extracellular Microvesicles from Cell Culture Medium: Comparative Evaluation of Methods. Biochem Suppl Ser B Biomed Chem. 2018;12: 167–175. doi:10.1134/S1990750818020117, ref. 19, 20, 32). 

SubXTM technology and line of isolation kits have been obviously developed over the past years. And now, an exosome isolation SubX reagent is also commercially available from Capital Biosciences (https://www.capitalbiosciences.com/exosome-isolation-kits). To our request to the company, we received the expected response that SubXTM is the trade secret of Capital Biosciences, as many commercially available kit components from different vendors. Information concerning SubXTM features are available on Capital Biosciences website (https://www.capitalbiosciences.com/liquid-biopsies ). But, as far as we know, there is no information in the scientific literature on the use of this kit for the isolation of extracellular vesicles. The studies presented in the manuscript demonstrate the effectiveness of the isolation of exosome-like vesicles by SubXTM reagent. This technology obviously has its advantages and disadvantages, some of which are considered in our study. We hope that the obtained data will be published and may become useful for research in the field of studying extracellular vesicles.

The comparative methods are all well known in the field. Many of the typical comparators are evaluated, though in the end it is not clear which metric sets this approach apart from the others. Indeed, there should have been a summary table inclusive of the metrics which were set out in the Abstract, “ . . efficacy, specificity, labor input, cost and scalability and standardization . .“. 

 It appears that the developed method may provide a high level of particle (assumed exosome) size homogeneity, but at the high expense of overall recoveries and much longer and complex processing protocols. 

>>We agree with these comments and, as suggested by the reviewer, we have included a comparison Table2 in the Discussion section of the revised manuscript.

As the method described here assumes to take advantage condensing vesicles together based on surface phosphate groups, the potential for co-condensation of phosphorylated proteins is never discussed nor experimentally ruled out. This is a fatal omission. Taken a step farther, a primary ‘contaminant’ in exosome isolation protocols is the inclusion of phospholipids. This point is never addressed.

>>Yes, indeed, the issue of contamination of extracellular vesicles with other plasma components, including proteins, is one of the main issues in the development of new approaches to isolating vesicles. However, contamination of vesicles with non-vesicular proteins should be taken into account in proteomic analysis of vesicular samples. However, the presence of additional proteins obviously does not affect the analysis of the miRNA profile of the isolated vesicles. It all depends on the ultimate goal of isolating exosomes from biological material. In this study, we convincingly demonstrate that using the SubXTM reagent, it is possible to isolate vesicles having a characteristic size (Fig. 2.5) and shape (Fig. 6) of exosomes, a bilipid membrane (Fig. 3) and exosomal markers on the surface (Fig. 7) (with complete absence of a negative marker of exosomes (Fig. 7)). A detailed study of the contamination of final vesicle samples, unfortunately, is beyond the scope of this study and is virtually impossible to implement using the methods used. The data obtained only allow us to assume the presence of minor protein contamination, based on the results of DLS shown in Figures 2 and 4. The corresponding minor additions were made to the Results section. In addition, available reagent descriptions indicate that for anchoring target molecule/microvesicle SubXTM requires oligophosphate cluster of at least 20 residues. This indirectly indicates the ineffectiveness of the SubXTM reagent to form complexes with proteins and phospholipids containing fewer phosphate groups, which in turn minimizes the potential for contamination of vesicles with phosphorylated molecules. According to the company website data SubXTM does not bind non-membrane integrated phosphoproteins and phospholipids molecules and aqueous micelles (https://www.capitalbiosciences.com/liquid-biopsies).

We also find the referencing to be sorely lacking as the series of recent papers from the group of Marcus et al., wherein polymer fibers are used with very high recoveries and purities, low-cost materials, and <15 min processing times for human plasma is ignored. 

>>Thank you for drawing our attention to the very interesting works of the Marcus group. In the revised manuscript, we quoted their data in the Introduction section (Ref.14: Bruce TF, Slonecki TJ, Wang L, Huang S, Powell RR, Marcus RK. Exosome isolation and purification via hydrophobic interaction chromatography using a polyester, capillary-channeled polymer fiber phase. Electrophoresis. 2019;40(4):571-581. doi:10.1002/elps.201800417). In our paper we did not discuss published, patented, and on-marked dominated group of “non-specific” methods and kits. Instead, we have compared three “affinity” techniques (immuno-, glicosacharide-, and oligophosphate-targeted) for exosome isolation along with two “golden standard” ultracentrifugation ones. 

Our more specific comments regarding the manuscript and methods follow (noted with respect to line number).

50. Should be the plural (EVs) 

>>corrected 

58. The last sentence of the paragraph should include references. 

>>Done 

70. While methods based on “hydrophobicity” are listed, none of the references relates to that approach. 

>>Thank you for your fair comment, we have added the appropriate reference.

79. We understand the intent, but is “disaggregate” a real term? “Dissociate” is probably more correct.

>>corrected 

82-84. These methods should not be capitalized. 

>>corrected 

99. What approach was used to isolate the plasma? Referencing. 

>>A brief description of the procedure is included in the Materials and Methods section. 

115. Is deuterated water (D2O) really used? If so, why? 

>>Yes, indeed we used deuterium water to prepare the sucrose cushion (30% sucrose/Tris/D2O, pH 7.4) exactly following the method detailed in ref. 17 of the revised manuscript.

116. The phrase “a fraction of pure vesicles” is totally vague. What size fraction? How “pure” are the vesicles? Pure is a very strong word.

>>corrected 

111-119. If this method used a sucrose cushion UC method and an ultra-filtration concentrator, how can you compare the recoveries of the "UC cushion technique" to the other isolation methods that did not employ this additional concentration step? You don't know if the EV recoveries are efficiently isolated by the UC sucrose cushion itself, or by the ultra-filtration concentrator. If this technique is to be truly compared to the others, the recoveries after the sucrose cushion UC method should be used for quantitative purposes, and if still necessary, the ultra-filtration should be used as an additional isolation method, where these techniques are employed in tandem, then compared to others. 3 of the 5 presented methods mentioned used ultrafiltration techniques. How can you account for this with comparison to others? 

>>A sucrose cushion eliminates more contaminants, such as proteins nonspeciﬁcally associated with exosomes, or large protein aggregates, which are sedimented by centrifugation (Szatanek R, Baran J, Siedlar M, Baj-Krzyworzeka M. Isolation of extracellular vesicles: Determining the correct approach (Review). Int J Mol Med. 2015;36(1):11-17. doi:10.3892/ijmm.2015.2194). Isolation of vesicles by ultracentrifugation into a sucrose cushion requires an additional sample concentration procedure. Repeated ultracentrifugation is required according to the procedure described in reference 17. To minimize isolation time, we modified this technique using ultrafiltration at this step. 

127. Again, the previously mentioned concerns with ultrafiltration apply. 

>>Undoubtedly, the ultrafiltration procedure makes both positive (additional purification of the sample) and negative (additional loss of a certain amount of vesicles) contributions. But here we used ultrafiltration (Amicon-Ultra, 100 kDa, Millipore) for purification of isolated EVs from Con-A and concentration of the preparation.

130-133. There are not details about the methodology here. In order to compare practical aspects such as method complexity and turnaround time, these need to be included. 

>>A brief description of the technique is included in the Materials and Methods section. 

105-144. Based on the individual descriptions, all of the methods require multiple hours to perform. This is a key point. A comprehensive tabulation is necessary. 

>>A table comparing the different vesicle isolation techniques used in our study is now included in the Discussion section of the revised manuscript.

179. The term “quantitative” is used quite loosely here. The data reported is presented as a number of “events”, and not converted to real numbers of particles. 

>>corrected 

186. How did you ensure the quantitative removal free proteins before the lysation of the vesicles? This technique specifically quantifies total protein. How do you know that impure EV recoveries are not positively skewing the quantification of EVs? We consider this a major flaw in the characterization.

>>As rightly noted, we measured total protein in all samples, including non-vesicular contaminating proteins. The corresponding clarification was introduced into the text. 

190. While the study presented in this section is quite convincing, what is not mentioned in subsequent sections is the potential that non-specific complexation of proteins will be an issue here because they also contain phosphate groups. How do the authors account for that? 

>>The issue of contamination with phosphorylated proteins of vesicle samples isolated by the SubXTM reagent was discussed above. Based on the results and the methods used in this study, we can only make assumptions about the possibility of contamination of the final EV samples. Corresponding minor additions have been made to the Results section of the revised manuscript.

Figs 2 and 4. It should be pointed out that the DLS data are presented as relative populations on normalized scales, not absolute number densities. 

>>Yes, the dynamic light scattering method allows one to determine the presence of particles of a certain size in a suspension, but does not allow an accurate estimate of their number. Figures 2 and 4 reflect the contribution of the scattering particles of a certain size population. Corresponding changes have been made to the both figures and their legends.

Fig. 2. Do you have DLS data of the sample after the incubation and before centrifugation? Because maybe a lot of exosomes didn't resuspend and still in the aggregates. DLS data couldn't give you the concentration information of exosomes. 

>>DLS measurements of the sample after the incubation of diluted plasma with SubX and before centrifugation show the particle aggregation. We believe that it is more informative to represent the particle size distribution in the supernatant (2B) and resuspended pellet (2C) obtained after incubation with SubX and the first centrifugation at 10,000 g., as shown in Figure 2. 

The DLS data presented in Figure 2D are completely identical to the data obtained after the last centrifugation. These results demonstrate the effectiveness of 300 mM salt in breaking down the formed aggregates of vesicles.

237. The results of Fig. 2A clearly shows how the quantification of EVs based on Bradford assays can be skewed by the presence of free protein and protein aggregates.

>>Figure 2А shows the particle size distribution in the blood plasma pre-cleaned from cell debris. Indeed, this sample contains a significant amount of plasma proteins that can potentially contaminate extracellular vesicles isolated from this pre-cleaned plasma during further isolation procedures. But the presence of free protein in the initial plasma sample does not at all correlate with its presence in the final samples of isolated vesicles. It is possible that the legend for panel A of figure 2 was not clear. The revised manuscript has been modified accordingly.

Fig. 3. Why are there figures A and B? What is the difference? 

>>corrected 

Fig. 5E. Mislabeled as “IB”. 

>>corrected

Fig. 6. The order of sub-figure items is presented opposite of the other figures, causing confusion.

>>corrected 

347. Potentially total protein could be used to quantify EVs, but you have to account for free protein contaminants. 

>>One of the most used techniques to quantify the EVs is the quantification of the total proteins of the isolated samples of EVs [17]. But indeed, protein measurements of EV-containing samples are inadequate to quantify EVs as pellets from a high-speed spin or isolated by any other methods may contain protein complexes/aggregates, lipoprotein particles, and other contaminants. However, the quantification of the total proteins of the isolated samples of EVs is an additional characteristic of the final sample of extracellular vesicles, along with other methods. In our opinion, the use of this approach in a comparative assessment of the effectiveness of several isolation methods is quite appropriate in addition to other methods of characterizing isolated vesicles used in the study.

402. The recoveries of the IP method are the same as the proposed method, so this needs to be noted. 

>>Exosome elution from IP beads was performed by incubation with Exosome Elution Buffer (Lonza), but [oligoEVs-SubXTM] complexes were dissociated back to monomers in the reconstruction buffer (300 mM NaCl).

421. Should “adventures” be “advantages”? 

>>corrected

---

## [Decision Letter · Decision Letter 1]

9 Nov 2020

Evaluation of immune and chemical precipitation methods for plasma exosome isolation

PONE-D-20-14332R1

Dear Dr. Shtam,

We’re pleased to inform you that your manuscript has been judged scientifically suitable for publication and will be formally accepted for publication once it meets all outstanding technical requirements.

Kind regards,

Girijesh Kumar Patel, PhD

Academic Editor

PLOS ONE

Additional Editor Comments (optional):

Reviewers' comments:

Reviewer's Responses to Questions

**Comments to the Author**

1. If the authors have adequately addressed your comments raised in a previous round of review and you feel that this manuscript is now acceptable for publication, you may indicate that here to bypass the “Comments to the Author” section, enter your conflict of interest statement in the “Confidential to Editor” section, and submit your "Accept" recommendation.

Reviewer #1: All comments have been addressed

Reviewer #2: (No Response)

2. Is the manuscript technically sound, and do the data support the conclusions?

Reviewer #1: Yes

Reviewer #2: Yes

3. Has the statistical analysis been performed appropriately and rigorously? 

Reviewer #1: Yes

Reviewer #2: N/A

4. Have the authors made all data underlying the findings in their manuscript fully available?

Reviewer #1: Yes

Reviewer #2: Yes

5. Is the manuscript presented in an intelligible fashion and written in standard English?

Reviewer #1: Yes

Reviewer #2: Yes

6. Review Comments to the Author

Reviewer #1: (No Response)

Reviewer #2: In this manuscript, authors compared different methods for the isolation of exosomes from the plasma. This manuscript is well written with appropriate experiments. However, there are some concerns like-

1. Authors should provide explanation/evidence for selecting CD63 to isolate exosomes, since exosomes express so many other markers on their surface.

2. Figure legend is poorly written, need an improvement.

3. Abstract should be finding focused, so should be revised as it contains so much background information.

7. PLOS authors have the option to publish the peer review history of their article (what does this mean?). If published, this will include your full peer review and any attached files.

Reviewer #1: No

Reviewer #2: No

---

## [Editor Report · Acceptance letter]

12 Nov 2020

PONE-D-20-14332R1 

Evaluation of immune and chemical precipitation methods for plasma exosome isolation 

Dear Dr. Shtam:

I'm pleased to inform you that your manuscript has been deemed suitable for publication in PLOS ONE. Congratulations! Your manuscript is now with our production department. 

Kind regards, 

on behalf of

Dr. Girijesh Kumar Patel 

Academic Editor

PLOS ONE